Differentially expressed serum proteins in children with or without asthma as determined using isobaric tags for relative and absolute quantitation proteomics

Li Ming 1
Wu Mingzhu 2
Qin Ying 3
Liu Huaqing 1
Tu Chengcheng 2
Shen Bing 3
Xu Xiaohong xuxiao1234@163.com 4
Chen Hongbo chenhongbo@ahmu.edu.cn 2
1 Department of Neonatology, Maternal and Child Health Hospital, the Affiliated Hospital of Anhui Medical University , Hefei , Anhui , China
2 Department of Obstetrics and Gynecology, Maternal and Child Health Hospital, the Affiliated Hospital of Anhui Medical University , Hefei , Anhui , China
3 School of Basic Medicine, Anhui Medical University , Hefei , Anhui , China
4 Department of Clinical Laboratory, Maternal and Child Health Hospital, the Affiliated Hospital of Anhui Medical University , Hefei , Anhui , China
Maddi Abhiram
Electronic publication date: 2020 Nov 3
Publication date: 2020
Volume: 8
Electronic Location ID: e9971
Received 2020 Apr 3; Accepted 2020 Aug 26
Copyright: ©2020 Li et al.
Copyright year: 2020
Copyright holder: Li et al.
License: This is an open access article distributed under the terms of the Creative Commons Attribution License, which permits unrestricted use, distribution, reproduction and adaptation in any medium and for any purpose provided that it is properly attributed. For attribution, the original author(s), title, publication source (PeerJ) and either DOI or URL of the article must be cited.
License URL: https://creativecommons.org/licenses/by/4.0/

Keywords: Children with asthma, Pathogenesis, Proteomics, iTRAQ, LC-MC/MS

Funding: Anhui Province Science and Technology Innovation Project Demonstration Project 201707d08050003 Anhui Province Key Research and Development Project 201904a07020032 National Natural Science Foundation of China 81570403 81371284 U1732157 This work was supported by grants from the Anhui Province Science and Technology Innovation Project Demonstration Project (No. 201707d08050003), and the Anhui Province Key Research and Development Project (No. 201904a07020032), and the National Natural Science Foundation of China (grant Nos. 81570403, 81371284 and U1732157). The funders had no role in study design, data collection and analysis, decision to publish, or preparation of the manuscript.

==============================
Background

Although asthma is one of the most common chronic, noncommunicable diseases worldwide, the pathogenesis of childhood asthma is not yet clear. Genetic factors and environmental factors may lead to airway immune-inflammation responses and an imbalance of airway nerve regulation. The aim of the present study was to determine which serum proteins are differentially expressed between children with or without asthma and to ascertain the potential roles that these differentially expressed proteins (DEPs) may play in the pathogenesis of childhood asthma.

Methods

Serum samples derived from four children with asthma and four children without asthma were collected. The DEPs were identified by using isobaric tags for relative and absolute quantitation (iTRAQ) combined with liquid chromatography tandem mass spectrometry (LC-MS/MS) analyses. Using biological information technology, including Gene Ontology (GO), Kyoto Encyclopedia of Genes and Genomes (KEGG), and Cluster of Orthologous Groups of Proteins (COG) databases and analyses, we determined the biological processes associated with these DEPs. Key protein glucose-6-phosphate dehydrogenase (G6PD) was verified by enzyme linked immunosorbent assay (ELISA).

Results

We found 46 DEPs in serum samples of children with asthma vs. children without asthma. Among these DEPs, 12 proteins were significantly (>1.5 fold change) upregulated and 34 proteins were downregulated. The results of GO analyses showed that the DEPs were mainly involved in binding, the immune system, or responding to stimuli or were part of a cellular anatomical entity. In the KEGG signaling pathway analysis, most of the downregulated DEPs were associated with cardiomyopathy, phagosomes, viral infections, and regulation of the actin cytoskeleton. The results of a COG analysis showed that the DEPs were primarily involved in signal transduction mechanisms and posttranslational modifications. These DEPs were associated with and may play important roles in the immune response, the inflammatory response, extracellular matrix degradation, and the nervous system. The downregulated of G6PD in the asthma group was confirmed using ELISA experiment.

Conclusion

After bioinformatics analyses, we found numerous DEPs that may play important roles in the pathogenesis of childhood asthma. Those proteins may be novel biomarkers of childhood asthma and may provide new clues for the early clinical diagnosis and treatment of childhood asthma.

Introduction

Asthma is one of the most common chronic noncommunicable diseases. According to reports from different countries, the current global prevalence rate of asthma is 1%–18% (Neelamegan et al., 2016), threatening approximately 334 million people worldwide. Asthma is characterized by airway hyperresponsiveness, reversible airway obstruction, and chronic airway inflammation. Symptoms are often recurrent and worsen with time (Papi et al., 2018). Asthma presents its highest incidence in childhood and affects children’s quality of life. Serious cases may lead to death, which brings great economic burdens to families and to health systems (Pincheira, Bacharier & Castro-Rodriguez, 2020). Therefore, identifying pathological mechanisms and finding new therapeutic targets for asthma are urgent.

Despite a worldwide presence and economic burden, the pathogenesis of childhood asthma is still unclear. It is currently thought that under the influence of genetic and environmental factors, the mechanisms underlying asthma include inflammatory cells, cytokines, and inflammatory mediators acting on the airway to cause airway inflammation and remodeling, and the imbalance of airway nerve regulation and abnormal structure and function of airway smooth muscle lead to airway hyperresponsiveness and induce asthma (Demenais et al., 2018; Morales & Duffy, 2019; Papi et al., 2018). There are two types of asthma. The first is called eosinophilic asthma and is characterized by an imbalance in the helper T (Th) cell Th1/Th2 ratio. The second type is called non-eosinophilic asthma and is mainly neutrophilic asthma that is controlled by Th17 (Lambrecht & Hammad, 2015). Children with asthma typically present with eosinophilic asthma and allergy, which easily leads to airway remodeling (Hamsten et al., 2016).

In asthma, some specific proteins produced by tissue cells may be secreted into the circulation. Thus, proteomics may be a useful approach to detect and quantitate such serum proteins and to determine whether they are differentially expressed in asthma and may be potential therapeutic targets. For example, by combining affinity proteomics with the human protein atlas, Hamsten et al. (2016) discovered that selective chemokine (C-C motif) ligand 5 (CCL5), hematopoietic prostaglandin D synthase (HPGDS), and neuropeptide S receptor 1 (NPSR1) were involved in inflammatory reactions and affected the onset of childhood asthma. In addition, Suojalehto et al. (2015) identified differentially regulated proteins by using two-dimensional differential gel electrophoresis and mass spectrometry. They determined that fatty acid binding protein 5 (FABP5) was increased in the sputum of patients with allergic asthma and showed the relationship of this protein with airway remodeling and inflammation (Suojalehto et al., 2015). However, among the currently available proteomics methods, isobaric tags for relative and absolute quantitation (iTRAQ) is considered to be most effective (Moulder et al., 2018). Therefore, to gain mechanistic insights into the pathogenesis of childhood asthma, the present study used iTRAQ technology combined with liquid chromatography tandem mass spectrometry (LC-MS/MS) to analyze the protein composition and expression levels in serum samples obtain from children with or without asthma. Using bioinformatics analyses, we aimed to determine key proteins that may be (1) used as biological markers or (2) part of critical signaling pathways involved in the development of asthma or (3) useful in determining the prognosis of children with asthma.

Materials & Methods

Experimental design

The study proceeded according to the flowchart shown in Fig. 1.

Figure 1 Flowchart of the experimental design.

COG represents Clusters of Orthologous Groups; GO, Gene Ontology; KEGG, Kyoto Encyclopedia of Genes and Genomes; iTRAQ, isobaric tags for relative and absolute quantitation; and SDS-PAGE, sodium dodecyl sulfate–polyacrylamide gel electrophoresis.

Clinical information and serum sample collection

From September to October 2019, serum samples were collected from eight children who had received no treatment but who had been admitted to the Maternal and Child Health Hospital of Anhui Medical University Affiliated Hospital. No child included in the study had received a diagnosis of an immune disease, chronic kidney disease, or other disease affecting serum proteins. The included samples were collected from four children with asthma (experimental group, all samples were belongs to eosinophilic asthma, and obtained after the diagnosis immediately but before drug treatment) and four children without asthma (control group). All clinical diagnoses followed the 2019 Global Initiative for Asthma guidelines. The experiment was approved by the Medical Ethics Committee of Anhui Medical University (approval number 20200284), and the parents or guardians of all participants signed informed consent forms.

On the morning of the second day after the children were admitted to the hospital without drug treatment, blood samples (four mL) were collected, placed at room temperature (22–25 °C) in the dark for 1 h, and centrifuged at 3000 rpm at 4 °C for 15 min. The supernatant was transferred to a new tube with a pipette and stored at −80 °C until it was used in an experiment.

Protein extraction and quality control

A ProteoExtract albumin/IgG removal kit (Merck & Co.) was used to extract the serum samples. The total amount of protein extracted from each one serum sample was more than 400 µg. The protein bands were clear, complete and uniform. The protein was not degraded, and the total amount of protein in each sample was able to be used to do two or more experiments. The protein solution was prepared, and the Bradford Protein Assay working solution was added. Sodium dodecyl sulfate–polyacrylamide gel electrophoresis (SDS–PAGE) and Coomassie blue staining were performed to evaluate the sample quality after detecting the total amount of protein.

Labeling after enzymatic hydrolysis of proteins

We used iTRAQ techniques to label the peptide segments after enzymolysis (Sandberg et al., 2012). After protein quantification, 60 µg of the protein solution was placed in a centrifuge tube and 5 µL dithiothreitol solution was added. The mixture remained at 37 °C for 1 h. Next, 20 µL of iodoacetamide solution was added, and the mixture was placed at room temperature in the dark for 1 h. The samples were centrifuged, and the collected supernatant was discarded. The precipitated sediment was twice treated with 100 µL of uric acid buffer (8 M urea, 100 mM Tris–HCl; pH 8.0). The samples were washed with NH4HCO3 (50 mM, 100 µL) three times, and trypsin (the ratio of protein to enzyme 50:1) was added to the sample in an ultrafiltration tube. Enzymolysis was performed on the samples at 37 °C for 12–16 h. Finally, the samples were labeled and desalted using a C18 cartridge.

LC-MS/MS analysis

We identified differentially expressed proteins (DEPs) between the two groups by using an LC-MS/MS Spectrum system (Sandberg et al., 2012). After re-dissolving the labeled samples in 40 µL of 0.1% formic acid aqueous solution, we analyzed them using nano-LC-MS/MS. The mobile phases were phase A (2% acetonitrile/0.1% formic acid/98% water) and phase B (80% acetonitrile/0.08% formic acid/20% water). The column was equilibrated with 95% phase A liquid. The gradient from phase B was adjusted as follows: 0–80 min, linear increase from 0% to 40%; 80–80.1 min, increased to 95%; 80.1–85 min, maintained at 95%; 85–88 min, decreased to 6%. The separated samples were analyzed using mass spectrometry with a Q Exactive HF-X Mass Spectrometer (Thermo Fisher Scientific). Proteome Discover software, version 2.2.0.388, was used to search the Uniport Human database in FASTA format. Protein fold changes (FC) of at least 1.5 were obtained, and P < 0.05 was considered a statistically significant difference. A FC < 1.5 represented upregulated proteins and a FC < 0.667 represented downregulated proteins; a FC between 0.667 and 1.5 indicated no obvious change in protein expression between the two groups (Morrissey et al., 2013; Oliveira et al., 2020; Rawat et al., 2016).

Gene Ontology (GO), Kyoto Encyclopedia of Genes and Genomes (KEGG), and Cluster of Orthologous Groups of Proteins (COG) Signaling Pathway Analyses

GO is a database that can be applied to various species to define and describe the functions of genes and proteins (Fang et al., 2019). The GO database is often used to clarify the roles of eukaryotic genes and proteins in cells. GO is useful for comprehensively describing the attributes of genes and gene products in organisms. GO consists of three domains (Liu et al., 2020; Wang et al., 2019a; Xing et al., 2020): (1) the cellular component domain contains the descriptions of proteins related to cell composition, which may be a subcellular structure (e.g., endoplasmic reticulum or nucleus) or a protein production component (e.g., ribosome, or proteasome); (2) the molecular function domain contains descriptions of all proteins related to molecular functions, such as biological activities and operations performed by specific gene products (i.e., molecules or complexes); (3) the biological process domain contains descriptions of all proteins related to biological processes, that is, a series of events in which molecular functions cooperate with one another. The GO function annotation result refers to the statistical number of DEPs detected between the two groups of serum samples in the three GO domains. The GO functional significance enrichment analysis provides the GO functional terms significantly enriched with the DEPs compared with all identified proteins, thus determining which biological functions the DEPs are significantly related to. The GO terms can explain the role of eukaryotic genes and proteins in cells, thus comprehensively describing the attributes of genes and gene products in organisms (Cai et al., 2015).

KEGG is a group of databases used to connect a series of genes in the genome with a molecular interaction network in cells to identify biological functions at the genomic and molecular levels. KEGG contains the signaling pathways of multiple cell processes, such as metabolism, membrane conversion, signal transduction, and the cell cycle (Kanehisa & Goto, 2000). The results of a KEGG analysis provides insights into the higher biological functions of cells. KEGG analysis can provide the most important biochemical metabolic pathways and signal transduction pathways involved in protein (Yang et al., 2018). COG annotates the functions of homologous proteins and includes both the COG database (clusters of homologous proteins from prokaryotes) and the KOG database (clusters of homologous proteins in eukaryotes) (Tatusov, Koonin & Lipman, 1997). COG database can provide the function of differential proteins (Wu et al., 2019). Briefly, using BLASTP (BLAST version 2.2.30, http://blast.ncbi.nlm.nih.gov/Blast.cgi), the sequence of protein set was compared with the COG database (the expected value of BLAST alignment parameter was set to 1 e-5), and the corresponding COG number of protein was obtained. The corresponding function description and function classification of protein were acquired.

Enzyme linked immunosorbent assay (ELISA)

Glucose-6-phosphate dehydrogenase (G6PD), the downregulated protein in the asthma group, was verified by ELISA kit. The experiment was performed according to the kit instructions. Blood samples were collected from children with or without asthma.

Statistical analysis

Two-tailed Mann–Whitney test was performed with SigmaPlot software. The chi-square test by Fisher’s exact test was used to compare the categorical variables (only gender data). Values are expressed as means ± SEM. A value of P < 0.05 was considered statistically significant.

Results

Participants

In the participants of LC-MS/MS experiments, no significant difference in age (asthma: control, means ± SD, 4.0 ± 1.0 vs. 3.3 ± 0.6 years; n = 8, P = 0.5, gender (asthma: control, 3 males and 1 female vs. 2 males and 2 females, P = 0.5) or body mass index (asthma: control, means ± SD, 19.1 ± 0.6 vs. 17.9 ± 0.9 kg/m2; n = 8, P = 0.3) was detected between the experimental group (children with asthma) and the control group (children with trauma but no infection or asthma).

SDS-PAGE

Serum samples obtained from children with or without asthma were separated using SDS-PAGE. Total protein from eight samples was effectively separated without protein degradation within the molecular weight range of 15–220 kDa. The protein levels were sufficient to be used in subsequent experiments (Fig. 2).

Figure 2 Protein separation and quality control using sodium dodecyl sulfate–polyacrylamide gel electrophoresis.

In this representative western blot, lane B1 is the original serum from a control child without asthma and M is the marker of B1. Lanes 1–4 are low-abundance proteins obtained from serum samples of children in the asthmatic group; and lanes 5–8 are low-abundance proteins obtained from serum samples of children in the control group.

LC-MS/MS analysis and identification of DEPs

LC-MS/MS is a powerful tool that enables identification of proteins in a mixed sample. In total, 103 proteins were identified in the serum of children with or without asthma, of which 46 DEPs were detected. As shown in Table 1, 12 DEPs were upregulated and 34 DEPs were downregulated. We plotted the magnitude of FC (log2 FC) on the x-axis and the statistical significance of that change (−log10 of the P value) on the y-axis to obtain a volcano plot (Fig. 3A). The cluster analysis for the expression of the DEPs clearly indicated that the expression patterns between children with or without asthma differed and that the protein expression in each group was clustered together (Fig. 3B). These results suggested that there was a significant difference in the levels of proteins expressed in the serum of children with or without asthma.

Table 1 Analysis of differential expressed proteins.

Accession indicates the characteristic numbers of different proteins. Description represents a detailed description of the proteins.

Regulated	Accession	Gene name	Description	FC	P	
UP	A0A087WW87	IGKV2-40	Immunoglobulin kappa variable 2-40	3.75	0.024	
UP	A0A193CHR8	–	10E8 light chain variable region (Fragment)	3.55	0.038	
UP	A2MYE1	–	A30 (Fragment)	3.23	0.006	
UP	A0A0B4J1X5	IGHV3-74	Immunoglobulin heavy variable 3-74	3.14	0.046	
UP	Q9UL81	–	Myosin-reactive immunoglobulin light chain variable region (Fragment)	3.1	0.007	
UP	A0A075B6S5	IGKV1-27	Immunoglobulin kappa variable 1-27	2.79	0.007	
UP	A2MYD2	V1-19	V1-19 protein (Fragment)	2.62	0.047	
UP	A0M8Q6	IGLC-7	Immunoglobulin lambda constant 7	2.45	0.002	
UP	A0A218KGR2	APP	Amyloid beta A4 protein isoform a	1.93	0.023	
UP	A2N7P4	–	Immunoglobulin mu-chain D-J4-region (Fragment)	1.84	0.041	
UP	P04430	IGKV1-16	Immunoglobulin kappa variable 1-16	1.83	0.031	
UP	P12273	PIP	Prolactin-inducible protein	1.75	0.038	
DOWN	C9JF17	APOD	Apolipoprotein D (Fragment)	0.67	0.049	
DOWN	H0YMF1	ACAN	Aggrecan core protein	0.66	0.003	
DOWN	Q12860	CNTN1	Contactin-1	0.62	0.020	
DOWN	C9IZP8	C1S	Complement C1s subcomponent (Fragment)	0.61	0.043	
DOWN	Q16853	AOC3	Membrane primary amine oxidase	0.6	0.025	
DOWN	Q07954	LRP1	Prolow-density lipoprotein receptor-related protein 1	0.59	0.030	
DOWN	Q03692	COL10A1	Collagen alpha-1(X) chain	0.58	0.001	
DOWN	P05556	ITGB1	Integrin beta-1	0.58	0.001	
DOWN	A8K6K4	–	cDNA FLJ77565, highly similar to Homo sapiens interleukin 1 receptor accessory protein (IL1RAP), transcript variant 1, mRNA	0.57	0.019	
DOWN	P23470	PTPRG	Receptor-type tyrosine-protein phosphatase gamma	0.57	0.049	
DOWN	Q76LX8	ADAMTS13	A disintegrin and metalloproteinase with thrombospondin motifs 13	0.57	0.043	
DOWN	P27487	DPP4	Dipeptidyl peptidase 4	0.56	0.026	
DOWN	P17301	IFNa 2	Integrin alpha-2	0.55	0.022	
DOWN	Q59EJ3	HSPA1A	Heat shock 70kDa protein 1A variant (Fragment)	0.55	0.033	
DOWN	Q59HB3	APOB	Apolipoprotein B variant (Fragment)	0.51	0.040	
DOWN	B4DWA6	–	cDNA FLJ60094, highly similar to F-actin capping protein subunit beta	0.5	0.041	
DOWN	O15394	NCAM2	Neural cell adhesion molecule 2	0.48	0.025	
DOWN	A0A3B3ISX9	TNXB	Tenascin-X	0.48	0.031	
DOWN	P60709	ACTB	Actin, cytoplasmic 1	0.47	0.007	
DOWN	Q9UIU0	CACNA2D1	Dihydropyridine receptor alpha 2 subunit	0.46	0.026	
DOWN	P30740	SERPINB1	Leukocyte elastase inhibitor	0.44	0.001	
DOWN	Q15063	POSTN	Periostin	0.44	0.039	
DOWN	A0A024RAG0	ALP	Alkaline phosphatase	0.4	0.029	
DOWN	A8K3K1	–	cDNA FLJ78096, highly similar to Homo sapiens actin, alpha, cardiac muscle (ACTC), mRNA	0.39	0.006	
DOWN	A0A024R5Z9	PK	Pyruvate kinase	0.39	0.021	
DOWN	A0A161I202	LTF	Lactoferrin	0.38	0.004	
DOWN	A0A140GPP7	–	Prolyl endopeptidase FAP	0.33	0.011	
DOWN	Q59G88	–	Coronin (Fragment)	0.33	0.007	
DOWN	P08246	ELANE	Neutrophil elastase	0.32	0.047	
DOWN	P08311	CTSG	Cathepsin G	0.3	0.026	
DOWN	P05164	MPO	Myeloperoxidase	0.23	0.014	
DOWN	B2RDY9	–	Adenylyl cyclase-associated protein	0.23	0.020	
DOWN	A8K8D9	G6PD	Glucose-6-phosphate 1-dehydrogenase	0.19	0.003	
DOWN	P07737	PFN1	Profilin-1	0.13	0.030	

GO functional annotation and enrichment analysis

The GO functional annotation results of the DEPs between children with or without asthma are shown in Fig. 4. For the cellular component domain, 37 DEPs were mainly concentrated in the cellular anatomical term, among which the top 3 upregulated proteins were IGKV2-40, IGHV3-74, and IGKV1-27. For the molecular function domain, the role of 38 DEPs was mainly in binding, among which the top 3 upregulated proteins were IGKV2-40, IGHV3-74, and IGKV1-27. For the biological process domain, the DEPS were primarily involved in cellular processes (31 DEPs) and responding to stimuli (30 DEPs), with the top 3 upregulated proteins being IGKV2-40, IGHV3-74, and V1-19.

Significant enrichment analysis of the GO function refers to a rough understanding of the biological processes in which DEPs are enriched based on the simple annotation of genes, which increases the reliability of research focused on determining pathogenesis. By analyzing the GO functional significance enrichment map of the DEPs between children with or without asthma (Fig. 5), it was found that the detected DEPs play important roles as cellular anatomical entities and in binding, cellular processes, and responding to stimuli.

KEGG metabolic pathway analysis

In organisms, different proteins perform their biological functions in coordination with one another. Analyses based on metabolic pathways are helpful to further understand the biological functions of the DEPs. KEGG is the main public database used to analyze such pathways, and analyses using KEGG can determine the most important biochemical metabolic pathways and signal transduction pathways in which the DEGs participate. Here, KEGG functional annotation analysis was carried out on the DEPs identified in the serum samples from children with or without asthma. The results showed that KEGG pathways annotated with the DEPs included dilated cardiomyopathy (DCM), hypertrophic cardiomyopathy (HCM), phagosomes, regulation of the actin cytoskeleton, focal adhesions, and metabolic pathways (Fig. 6). The downregulated DEPs associated with DCM/HCM were ITGB1, ITGA2, ACTB, and CACNA2D1. The downregulated DEPs associated with phagosomes were ITGB1, ITGA2, ACTB, and MPO. The downregulated DEPs associated with regulation of the actin cytoskeleton were ITGB1, ITGA2, ACTB, and PFN1. The downregulated DEPs associated with focal adhesions were ITGB1, ITGA2, TNXB, and ACTB. The downregulated DEPs associated with metabolic pathways were AOC3, ALPL, PKM2, and G6PD. The pathway enrichment analysis is the same as the GO functional enrichment analysis, which uses the KEGG pathway as a unit and applies hypergeometric testing to determine the pathways significantly enriched with DEPs compared with all identified proteins. The most important biochemical metabolic pathways and signal transduction pathways of DEPs can be determined by pathway enrichment analysis (Fig. 7). The present analysis showed that most of the detected downregulated proteins were concentrated in metabolic pathways such as focal adhesion, MAPK signaling pathway, platelet activation and Rap1 signaling pathway.

COG protein functional analysis

COG is a database for orthologous classification of proteins. It is assumed that the group of proteins constituting each COG are all derived from the same ancestral protein and are divided into orthologs and paralogs. Orthologs refer to proteins evolved from vertical families (speciation) of different species and typically retain the same functions as the original proteins. Paralogs refer to proteins derived from gene replication in certain species that may evolve new functions related to the original functions. The DEPs between children with or without asthma were analyzed using the COG database (Fig. 8). We predicted the possible functions of these protein, and generated the functional classification statistics. The results showed that the functions of these DEPs were mainly concentrated in signal transduction mechanisms; posttranslational modification, protein turnover, and chaperones; amino acid transport and metabolism; cytoskeleton; and extracellular structures.

Figure 3 Volcano plot and heatmap of differentially expressed proteins (DEPs) in children with or without asthma.

(A) In this volcano plot, yellow dots represent proteins with a significant fold change (FC) >1.5; blue-green dots, proteins with a significant FC < 0.667; black dots, proteins with no significant change. (B) In a heatmap, the upregulation and downregulation of different proteins are observed by cluster analysis. Each line in the figure represents a protein, each column is a sample (C1-4: the asthmatic group, E1-4: the control group), and the colors represent different expression levels (the log2 value of the quantitative value is obtained and a median correction is carried out during drawing of the heatmap).

Figure 4 Gene Ontology (GO) annotation for functional classification.

The x-axis represents the classification description (GO term) grouped under each of the three GO domains listed above the plot, and the y-axis represents the number of differentially expressed proteins for each of these terms.

Figure 5 Gene Ontology (GO) enrichment for functional terms.

The x-axis represents the enrichment factor, that is, the proportion of differentially expressed proteins in the total proportion of this GO classification relative to a multiple of the total proportion of differentially expressed proteins in the GO classification relative to the proportion of identified proteins in the classification. The y-axis provides the GO term descriptions. The size of the circle represents the number of differentially expressed proteins in the GO term, and the color of the circle indicates the statistical significance of the finding. Fisher exact test P value: P value of the enrichment test obtained using the Fisher exact test; −log10 (P value): log conversion of the Fisher exact test P value.

Figure 6 Kyoto Encyclopedia of Genes and Genomes (KEGG) pathway annotation.

The horizontal axis represents the number of differentially expressed proteins (DEPs), and the vertical axis provides the description of each KEGG pathway. Thus, the number of DEPs identified in this experiment is shown for each specific KEGG signaling pathway annotated for those proteins.

Figure 7 Kyoto Encyclopedia of Genes and Genomes (KEGG) pathway enrichment.

The horizontal axis represents the enrichment factor, that is, the total proportion of differentially expressed proteins in the KEGG signaling pathway is a multiple of the change in the proportion of identified proteins in the classification. The vertical axis provides a description of the KEGG classification. The size of the circle represents the number of differentially expressed proteins in the KEGG signaling pathway, and the color of the circle indicates the statistical significance of the finding. Fisher exact test P value: P value of the enrichment test obtained using the Fisher exact test; −log10 (P value): log conversion of the Fisher exact test P value.

Figure 8 Clusters of Orthologous Groups (COGs) annotated for differentially expressed proteins.

The horizontal axis represents the functional code in the COG database, and the description of that code is shown on the right side of the figure. The y-axis indicates the frequency of each functional code.

G6PD protein concentration in serum

To confirm the DEP in the LC-MS/MS experiment, we used an ELISA experiment and choose a key protein G6PD, which is decreased in asthma reported by studies from other groups (Hirasawa et al., 2003). Our data also showed a significant reduction in asthma group compared to control group (Fig. 9). Therefore, the result is consistent with the data from LC-MS/MS experiment.

Figure 9 Glucose-6-Phosphate Dehydrogenase (G6PD) protein concentration in serum.

The horizontal axis represents the control group and asthma group. The y-axis indicates the concentration of G6PD. The proteins were extracted from the serum of children with or without asthma. Values are shown as the mean ± SEM (n = 8); ∗P < 0.05 for Control vs. Asthma.

Discussion

The pathological mechanisms underlying asthma are still not clear; however, the application of proteomics may be valuable for finding some new clues. Recently, several studies also used proteomics to try to find biomarkers in asthma. Gharib et al. (2011) utilized shortgun proteomics method to identify protein expression pattern in adult induced sputum samples; Bhowmik et al. (2019) used quantitative label-free liquid chromatography–tandem mass spectrometry method to find that apolipoprotein E (ApoE) is significantly downregulated but interleukin 33 (IL-33) is significantly upregulated in serum samples from adult atopic asthma compared to healthy control; Moreover, helped by method of iTRAQ combined with LC-MS/MS, Liu et al. (2017) reported DEPs in serum samples from children patients with controlled, partly controlled, or uncontrolled childhood asthma. However, in our study, we used iTRAQ method to identify the DEPs in the serum of children with asthma vs. those without asthma. Asthma is an allergic response, that is, an individual’s own immune system overreacts. In a previous study, it was found that when the human airway is exposed to invading pathogens, the congenital immune process is rapidly induced (Lebold, Jacoby & Drake, 2016). The congenital immune system is the first line of defense in the human immune system (Yin et al., 2020). When the body is infected by foreign agents, inflammatory reactions will occur first (Kimbrell & Beutler, 2001). These reactions can produce a variety of chemical factors, including cytokines (Kzhyshkowska & Bugert, 2016), to recruit immune cells (e.g., macrophages, neutrophils) to infected or inflamed tissues to kill or inhibit the growth of pathogens through phagocytosis and other actions to prevent the spread of infection. Inflammatory reactions can also promote the healing of injured tissues. Another defensive reaction to invading pathogens is the complement system, which helps or complements the antibody itself to remove or to label antigenic substances to control pathogens (Hato & Pierre, 2015). Complement component 1s (C1S) is involved in the complement and coagulation cascade in the metabolic pathway of the KEGG database. In our GO analysis, we found that several proteins that are involved in complement activation (namely, IGHV3-74, IGLC7, and IGKV1-16) were differentially expressed between children with or without asthma. These mass spectrometry results are consistent with results from a previous study (Hato & Pierre, 2015). However, we also detected DEPs that have not been reported in previous studies on childhood asthma, including IGKV2-40, IGHV3-74, IGKV1-27, IGKV1-16, CTSG, HSPA1A, SERPINB1, LTF, IGLC7, CTSG, ADAMTS13, NCAM2, TNXB, ACTB, and CNTN1. These findings may indicate that when infection causes a rapid congenital immune response and airway inflammation, airway remodeling will be induced, leading to the onset or exacerbation of childhood asthma (Krusche, Basse & Schaub, 2019; Johnston et al., 1995). In addition, inflammation is divided into local manifestations and systemic reactions. Small trauma may cause local inflammation. When local lesions are serious, especially when pathogenic microorganisms spread in the body, obvious systemic reactions often occur. The children in the control group had only minor local trauma which did not develop systemic inflammation. However, the inflammatory reaction in asthmatic children prefers a systemic inflammation.

By analyzing DEPs through GO terms or KEGG pathway analysis, we found that the upregulated proteins IGKV2-40, IGHV3-74, IGKV1-27, and IGKV1-16 were involved in the innate immune process. Most of the downregulated proteins were involved in the congenital immune system and neutrophil degranulation, including CTSG, MPO, IFNa2, HSPA1A, SERPINB1, C1S, and LTF. Among them, IFNa2 is produced after the body recognizes many pathogens and damage-related molecules released by infected or dead cells and is a key component of the innate immune response (Paul, Pellegrini & Uzé, 2015). HSPA1A can exert immune functions through fusion with the membrane, endocytosis, autophagy, and interaction with ligands (Bilog et al., 2019; Oliverio et al., 2018; Sangaphunchai, Todd & Fairclough, 2020). SERPINB1 is a neutrophil elastase inhibitor that plays an important role in regulating cell activity, inflammatory response, and cell migration (Torriglia, Martin & Jaadane, 2017). Therefore, our findings provided several new potential protein targets in the immune system for treatment of children with asthma.

Among the upregulated proteins, the GO terms or KEGG pathways analysis showed that the DEPs participate in immunoglobulin production (IGKV2-40, IGKV1-27, and IGKV1-16), receptor-mediated endocytosis (IGKV2-40, IGLC7, and IGKV1-16), phagocytosis, recognition and engulfment (IGHV3-74, IGLC7), and complement activation (IGHV3-74, IGLC7, and IGKV1-16). The downregulated proteins are involved in extracellular matrix degradation (CTSG, TNXB, ELANE, and COL10A1), metabolism of various substances (CTSG, ADAMTS13, DPP4, G6PD, APOB, ACAN, AOC3, and LRP1), and cell adhesion and focal adhesion (NCAM2, TNXB, ACTB, CNTN1, ITGB1, and ELANE). During airway remodeling, epithelial cell exfoliation, goblet cell proliferation, vascular proliferation, extracellular matrix deposition, and hypertrophy of smooth muscle cells are involved. When the expression levels of ELANE are downregulated, the degradation of the extracellular matrix decreases, which will aggravate airway remodeling (Linden, Laan & Anderson, 2005). Other group (Hirasawa et al., 2003) found that MAPK and activated protein kinases are related to viral replication. Lacking G6PD will increase the products of cellular reactive oxygen species, which strengthens those kinase pathways to facilitate viral replication, thus inducing or aggravating asthma. AOC3 is commonly referred to as vascular adhesion protein-1 (VAP-1) and is expressed in lymphocyte endothelial interactions. As a new marker of myofibroblasts, AOC3 may play a role in pulmonary fibrosis and thus induce asthma (Hsia et al., 2016). The imbalance in the expression levels of LRP1 in fibroblasts of healing tissues may lead to unlimited expansion of contractile fibroblasts, thus causing or aggravating pulmonary fibrosis and participating in the pathogenesis of asthma (Schnieder et al., 2019). Neural cell adhesion molecule 2 (NCAM2) protein has been shown to reduce inflammation in Alzheimer’s disease (Rasmussen et al., 2018). CNTN1, which is also a cell adhesion protein, can promote the invasion of prostate cancer cells, enhance Akt activation, and reduce the expression of epithelial cadherin in cancer cells (Yan et al., 2016). Some studies have found that DPP4 plays an important role in angiogenesis and growth, immune response, cell proliferation, fibrin degradation, cytokine production, signal transduction, and other physiological and pathological processes of the body (Liu et al., 2015; Mark, 2005; Ohnuma, Hatano & Morimoto, 2015; Ta et al., 2010). In addition, DPP4 is widely distributed in lung tissue and participates in pulmonary inflammation and the formation of pulmonary surfactant (Mentlein, 1999; Schmiedl et al., 2014). We speculate that when a pathogenic infection exists in the lungs of children, the downregulation of DPP4 will lead to an immune imbalance and inflammatory cascade reaction to affect the formation of lung substances, thus inducing asthma. However, our findings and speculation should be confirmed in future studies.

Other DEPs detected in our KEGG analysis also play important roles in cellular physiological and pathological activities and may be part of mechanisms underlying childhood asthma. Among the upregulated proteins, APP participates in the following pathways: neuron growth, adhesion, axon generation, cell migration, regulation of protein movement, cell apoptosis, and combination with extracellular matrix components. Some studies have also found that APP plays an important role in the pathogenesis of Alzheimer’s disease and is overexpressed in cancer cells, such as in nasopharyngeal carcinoma (Li et al., 2019a; Li et al., 2019b). Another upregulated protein, PIP, a pathway annotated in KEGG, has a regulatory effect on natural immune cells. Some studies have found that PIP can specifically degrade fibronectin to help hosts resist infection and play a role in the migration, adhesion, and invasion of cancer cells (Ihedioha et al., 2018; Kitano et al., 2006; Wang et al., 2019b).

Among the downregulated proteins detected through our GO term or KEGG pathway analysis, CACNA2D1 is involved in the macrophage CCR5 pathway; ITGB1 is involved in the adaptive immune process, cell adhesion, blood–brain barrier, immune cell migration, and cytoskeleton signaling process; and LTF is involved in amyloid fiber formation. Thus, through KEGG analyses, we have provided insights into the potential mechanisms involved in the development of childhood asthma.

Conclusions

We used iTRAQ combined with LC-MS/MS technologies to identify DEPs in serum samples between children with or without asthma. The present study provided additional evidence consistent with the role of the inflammation-immune mechanism in the pathogenesis of childhood asthma, and offered new potential biomarkers for childhood asthma that may be helpful for its early diagnosis. The analysis of signaling pathways provided numerous key proteins that may be involved in the development of childhood asthma and that may be new targets in the study of asthma in future.

Supplemental Information

Supplemental Information 1 Full length uncropped blots of Fig. 2

Click here for additional data file.

Supplemental Information 2 Raw measurements

In total, 103 proteins were identified in the serum of children with or without asthma, of which 46 DEPs were detected. As shown, 12 DEPs were upregulated and 34 DEPs were downregulated.

Click here for additional data file.

Additional Information and Declarations

Competing Interests

Author Contributions

Human Ethics

Data Availability

The authors declare there are no competing interests.

Ming Li and Mingzhu Wu conceived and designed the experiments, performed the experiments, analyzed the data, prepared figures and/or tables, authored or reviewed drafts of the paper, and approved the final draft.

Ying Qin conceived and designed the experiments, performed the experiments, prepared figures and/or tables, and approved the final draft.

Huaqing Liu conceived and designed the experiments, analyzed the data, authored or reviewed drafts of the paper, and approved the final draft.

Chengcheng Tu conceived and designed the experiments, performed the experiments, authored or reviewed drafts of the paper, and approved the final draft.

Bing Shen conceived and designed the experiments, prepared figures and/or tables, authored or reviewed drafts of the paper, and approved the final draft.

Xiaohong Xu and Hongbo Chen conceived and designed the experiments, authored or reviewed drafts of the paper, and approved the final draft.

The following information was supplied relating to ethical approvals (i.e., approving body and any reference numbers):

The experiment was approved by the Medical Ethics Committee of Anhui Medical University (approval number 20200284).

The following information was supplied regarding data availability:

The raw measurements are available in the Supplementary Files.

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
