# Peer review of "Differentially expressed serum proteins in children with or without asthma as determined using isobaric tags for relative and absolute quantitation proteomics"

_PeerJ, doi:10.7717/peerj.9971_

## Round 0.1 · original submission · Major Revisions

Please address the reviewer's comments on a point by point basis and submit your revised manuscript.

·

Basic reporting

Please have a look at Minor concerns: 1 to 3.

Experimental design

Please have a look at Major concerns: 1 to 5.

Validity of the findings

No comment.

Additional comments

Asthma is one of the most common chronic noncommunicable diseases, which is a chronic inflammatory disease of the lower airway that is commonly encountered by the otolaryngologist. Asthma is a serious global health issue and asthma guidelines recommend a stepwise approach to management with goals to achieve control and minimize future risk. It is a good idea to investigate the protein composition and expression levels in serum samples obtain from children with or without asthma. In this manuscript, Dr. Fang and colleagues looked at the role of the differentially expressed proteins (DEPs) identified by using isobaric tags for relative and absolute quantitation (iTRAQ) combined with liquid chromatography tandem mass spectrometry (LC-MS/MS) analyses in the pathogenesis of childhood asthma. The study provides the conclusion, which showed that numerous DEPs that may play important roles in the pathogenesis of childhood asthma. Those proteins may be novel biomarkers of childhood asthma and may provide new clues for the early clinical diagnosis and treatment of childhood asthma. Although, the current study is interesting and well but there are some major concerns that need to be addressed.

Major concerns:
1. “Experimental design & Validity of the findings” In general, the research idea of this manuscript is novelty. However, the biggest defect is the lack of sample quantity, which makes it difficult to judge whether the conclusion is universal or not. If the authors increase each group of samples to 10, where possible, the results should be more applicable. However, considering the high cost of iTRAQ technology, I recommend that the author select 1 or 2 from the 37 DEPs for ELISA verification, and expand the samples to about 50+, which may significantly improve the impact of this study.
2. “Materials & Methods – Clinical information and serum sample collection” Line 103-112. “The included samples were collected from four children with asthma (experimental group) and four children without asthma (control group).” When grouping, did the authors take into account factors such as age, gender, and body mass index (BMI) of patients in the balance between the two groups? Because the physiological factors mentioned above also have a certain influence on serum protein, considering the small sample size of this study, these factors must be matched to exclude the relevant influence, please explain in the article. P.S. I saw that the author showed no difference in age and gender between the two groups in the results section. Please analyze BMI also.
3. “Materials & Methods – LC-MS/MS analysis” Line 147-150. “Protein fold changes (FC) of at least 1.5 were obtained, and P < 0.05 was considered a statistically significant difference. A” Generally, in high-throughput analysis, the threshold value of FC is defined at 2. Why the author chooses 1.5, please clarify the reasons and references.
4. “Table 1” Page 45. The data showed that G6PD was one of the most significantly down-regulated proteins. This is a very interesting finding. As an important gene related to G6PD deficiency disease, the detection of G6PD activity should be widely used in children's hospitals and general hospitals in mainland China. Can the author make a retrospective study, collect and analyze relevant data, and observe whether the activity of G6PD is down-regulated in the blood of children with asthma? The research value of this paper can be further enhanced by appropriately expanding the sample size.
5. Through PubMed searching, the keywords are "asthma" and "iTRAQ", and I find that there are very few relevant studies. There is a lack of research on the detection of serum in children with asthma, so I think the authors should try to add as many samples as possible.

Minor comments:
1. Figure 3B. The author should indicate which group “B1-4” and “A1-4” belong to?
2. Figure 4. The text in the picture is too small. Please increase the font size.
3. All Figure. The vector image exported by R language can be modified with drawing software (Canvas or AI) again. Please choose the appropriate font size for the convenience of readers. In addition, author should keep the horizontal and vertical ratio in the picture zooms in and out.

Reviewer 2 ·

Basic reporting

No comment.

Experimental design

No comment.

Validity of the findings

No comment.

Additional comments

This manuscript by Ming Li found 46 DEPs in serum samples of children with asthma vs. children without asthma. Verified some proteins that have been reported by others, and also provided several new potential proteins. These newly discovered proteins maybe some alternative targets in future studies of childhood asthma.
Here are a few questions and comments:
1. Line 84-86 “……selective chemokine ligand 5, hematopoietic prostaglandin D synthase, and neuropeptide S receptor 1 were involved……” line 88 “……fatty acid binding protein 5……” and line 272 “Complement component 1s……” Do all these descriptions refer to specific genes? Please mark their official symbol accurately.
2. Line 269 mentions “Inflammatory reactions can also promote the healing of injured tissues.” In the results of the manuscript, lines 184-185, the description of the control group is “children with trauma ……” Will the trauma here affect the experimental results? Maybe some discussion needs to be added.
3. Figure 3A, please accurately describe or set the color of the volcano plot. The description in figure legends “red dots” and “green dots” should be “yellow dots” and “blue-green dots” respectively.
4. Figure 8 need to adjust because some have exceeded the maximum value of the Y-axis.
5. Although “……Children with asthma typically present with eosinophilic asthma and allergy……” (line 77), the pathology type of children with asthma should be provided so that readers can clearly understand the possible impact on the experimental results.
6. line 149 “……>1.5 represented upregulated proteins and a FC <0.667 represented downregulated proteins……” however, the description in Figure 3 is “FC> 1.5” and “FC <1.5” as a significant change. And the volcano uses log10FC, and the heatmap uses log2FC, please explain clearly.

·

Basic reporting

1. Several studies that tried to identify differentially expressed proteins of asthma have been published (as listed below) and one also focused on childhood asthma. Could the authors explain the difference and significance regarding the design and results of their study compared to those studies?
1) Pilot-Scale Study Of Human Plasma Proteomics Identifies ApoE And IL33 As Markers In Atopic Asthma. J Asthma Allergy. doi: 10.2147/JAA.S211569.
2) Screening Serum Differential Proteins for Childhood Asthma at Different Control Levels by Isobaric Tags for Relative and Absolute Quantification-based Proteomic Technology. Zhongguo Yi Xue Ke Xue Yuan Xue Bao. DOI: 10.3881/j.issn.1000-503X.2017.06.014
3) Induced Sputum Proteome in Healthy Subjects and Asthmatic Patients. J Allergy Clin Immunol. doi: 10.1016/j.jaci.2011.07.053.

2. Some of the references are old, and the authors should also refer to studies of top journals in the field.

Experimental design

1. The authors should describe the amount of protein they extracted from the blood sample as well as the amount they applied in each tests respectively.

2. The statistical analysis applied in this study should be mentioned in method.

3. In method, further explanations in terms of how the analyses of GO, KEGG, COG were carried out together with their meanings in the study should be described, instead of only focusing on their definitions.

Validity of the findings

No comments

Additional comments

In this study titled “Differentially expressed serum proteins in children with or without asthma as determined using isobaric tags for relative and absolute quantitation proteomics”, the author tried to differentially expressed proteins (DEPs) expressed between children with or without asthma. Among these DEPs in the result, 12 proteins were significantly upregulated and 34 proteins were downregulated. The author further analyzed these DEPs with GO, KEGG and COG to explain the meaning of identified DEPs in asthma.

However, several issues need to be addressed before further consideration for publication.

1. Serum samples derived from four children with asthma and four children without asthma were collected in this study, which is quite a small sample size compared with other similar studies. Could the author give their reason or evidence to support that this sample size is sufficient enough?

2. For those four children with asthma, were they hospitalized due to asthma exacerbation, or just with a history of chronic asthma? Furthermore, since their blood was collected the other day of their hospitalization, they were supposed to be treated with antiasthmatic drugs. Did the authors notice the impact of antiasthmatic drugs on DEPs, and the difference of DEPs during asthma exacerbation, which the authors should try to explain?

3. Should the authors explain the reasons and evidence why they chose to analyze the blood of the patients instead of sputum or samples derived from bronchoalveolar lavage?

---

## Round 0.2 · Minor Revisions

Please address the comments of the reviewers and submit a revised manuscript.

·

Basic reporting

No comment.

Experimental design

No comment.

Validity of the findings

No comment.

Additional comments

Asthma is one of the most common chronic noncommunicable diseases, which is a chronic inflammatory disease of the lower airway that is commonly encountered by the otolaryngologist. Asthma is a serious global health issue and asthma guidelines recommend a stepwise approach to management with goals to achieve control and minimize future risk. It is a good idea to investigate the protein composition and expression levels in serum samples obtain from children with or without asthma. In this manuscript, Dr. Fang and colleagues looked at the role of the differentially expressed proteins (DEPs) identified by using isobaric tags for relative and absolute quantitation (iTRAQ) combined with liquid chromatography tandem mass spectrometry (LC-MS/MS) analyses in the pathogenesis of childhood asthma. The study provides the conclusion, which showed that numerous DEPs that may play important roles in the pathogenesis of childhood asthma. Those proteins may be novel biomarkers of childhood asthma and may provide new clues for the early clinical diagnosis and treatment of childhood asthma. After the modification, the data quality and language description are improved. Although there are still some deficiencies (sample size is small) in this study, there are few relevant explorations in this field. This study is a good case worthy of publication and guidance for further research.

Reviewer 2 ·

Basic reporting

no comment

Experimental design

no comment

Validity of the findings

no comment

Additional comments

Some minor issues need to be corrected by the author.
1. Please describe the clustering method used in the cluster analysis in Materials and Methods.
2. CCL5 should be chemokine (C-C motif) ligand 5 or C-C motif chemokine ligand 5. Please read the references carefully.
3. The Official Symbol of Complement component 1s should be C1S. Please pay attention to the wording of gene names to distinguish different species.

---

## Round 0.3 · accepted · Accept

The changes you have made in the manuscript have improved the manuscript. Congratulations!!